# COLLABORATIVE FILTERING WITH SMOOTH RECONSTRUCTION OF THE PREFERENCE FUNCTION

## ABSTRACT

The problem of predicting the rating of a set of users to a set of items in a recommender system based on partial knowledge of the ratings is widely known as collaborative filtering. In this paper, we consider a mapping of the items into a vector space and study the prediction problem by assuming an underlying smooth preference function for each user, the quantization at each given vector yields the associated rating. To estimate the preference functions, we implicitly cluster the users with similar ratings to form dominant types. Next, we associate each dominant type with a smooth preference function; i.e., the function values for items with nearby vectors shall be close to each other. The latter is accomplished by a rich representation learning in a so called frequency domain. In this framework, we propose two approaches for learning user and item representations. First, we use an alternating optimization method in the spirit of $k$-means to cluster users and map items. We further make this approach less prone to overfitting by a boosting technique. Second, we present a feedforward neural network architecture consisting of interpretable layers which implicitely clusters the users. The performance of the method is evaluated on two benchmark datasets (ML-100k and ML-1M). Albeit the method benefits from simplicity, it shows a remarkable performance and opens a venue for future research. All codes are publicly available on the GitLab.

## 1 INTRODUCTION

Nowadays, recommender systems (RS) are among the most effective ways for large companies to attract more customers. A few statistics are sufficient to attract attention towards the importance of RS: 80 percent of watched movies on Netflix and 60 percent of video clicks on Youtube are linked with recommendations (Gomez-Uribe & Hunt, 2015; Davidson et al., 2010). However, the world of RS is not limited to video industry.

In general, recommender systems can be categorized into three groups (Zhang et al., 2019): collaborative filtering (CF), content-based RS, and hybrid RS depending on the used data type. In this paper, we focus on CF, which uses historical interactions to make recommendations. There might be some auxiliary information available to the CF algorithm (like the user personal information); however, a general CF method does not take such side information into account (Zhang & Chen, 2019). This includes our approach in this paper.

Recently, deep learning has found its way to RS and specifically CF methods. Deep networks are able to learn non-linear representations with powerful optimization tools, and their efficient implementations have made then promising CF approaches. However, a quick look at some pervasive deep networks in RS (e.g., He et al. (2017) and Wu et al. (2016)) shows that the utilization of deep architectures is limited to shallow networks. Still, it is unclear why networks have not gone deeper in RS in contrast to other fields like computer vision (Zhang et al., 2019). We suppose that the fundamental reason that limits the application of a deeper structure is the absence of interpretability (look at Seo et al. (2017), for example). Here, interpretability can be defined in two ways (Zhang et al., 2019); first, users be aware of the purpose behind a recommendation, and second, the system operator should know how manipulation of the system will affect the predictions (Zhang et al., 2018).

This paper addresses both issues by formulating the recommendation as a smooth reconstruction of user preferences. Particularly, our contributions are:

- The CF problem is formulated as the reconstruction of user preference functions by minimal assumptions.

- An alternating optimization method is proposed that effectively optimizes a non-convex loss function and extracts user and item representations. In this regard, effective clustering methods are proposed and tested.

- A feed-forward shallow architecture is introduced, which has interpretable layers and performs well in practice.

- Despite the simplicity and interpretability of the methods, their performance on benchmark datasets is remarkable.

## 1.1 RELATED WORKS

The applied methods in CF are versatile and difficult to name. Below, we explain a number of methods which are well-known and are more related to our work.

*Multilayer perceptron based models.* A natural extension of matrix factorization (MF) methods (Mnih & Salakhutdinov, 2008) are Neural Collaborative Filtering (NCF) (He et al., 2017) and Neural Network Matrix Factorization (NNMF) (Dziugaite & Roy, 2015). Both methods extend the idea behind MF and use the outputs of two networks as the user and the item representations. The inner-product makes the prediction of two representations. Although our work has some similarity to this method, we model users by functions and represent these functions in a so-called frequency domain. Thus, user and item representations are not in the same space.

*AutoEncoder based models.* AutoRec (Sedhain et al., 2015) and CFN (Strub et al., 2016) are well-known autoencoder (AE) structures that transform partial observations (user-based or item-based) into full row or column data. Our method differs from AE structures as our network use item (user) representations and predicts user (item) ratings.

## 2 SMOOTH RECONSTRUCTION FROM NON-UNIFORM SAMPLES

*Rating as the output of the preference function.* Most of the time, a finite set of features can characterize users and items that constitute the recommendation problem. Although no two users or items are exactly the same, the number of characterization features can be considerably small without losing much information.

Assume that item $i$ is characterized by the vector $\boldsymbol{x}_i \in \mathcal{X} \subset \mathbb{R}^d$. We further assume that all users observe similar features of an item and user $u$'s ratings are determined by a preference function $f_u : \mathcal{X} \to [c_{min}, c_{max}]$. The recovery of a general preference function might need an indefinite number of samples, i.e., observed ratings. However, we do not expect user attitudes to change too much with small changes in an item's feature. E.g., if the price is the determinative factor in someone's preference, small changes in the price must not change the preference over this item significantly (look at figure 1).

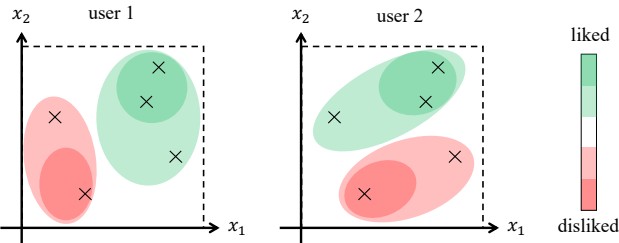

Figure 1: Preference function is expected to have smooth behavior over the space of items.

*Reconstruction of bandlimited 1D signals.* Let us start with the simplest case. Consider $s[n], n = 0, 1, \ldots, N-1$, a 1D signal with length $N$. We call $s$ to have bandwidth $M < \frac{N}{2}$ if there is a representation $\hat{s}[m], m = -M, -M+1, \ldots, M-1, M$ that can represent $s$ as:

$$s[n] = \sum_{m=-M}^{M} \widehat{s}[m] e^{j2\pi(mn/N)} \tag{1}$$

So $2M+1$ distinct samples from $s$ would be enough to calculate $\widehat{s}$. For an analytical approach, it is useful to interpret equation 1 as a discretization of a trigonometric continuous equation:

$$h(x) = \sum_{(m=-M)}^{M} a_m e^{j2\pi(mx)}, \; \boldsymbol{a} \in \mathbb{C}^{2M+1}, \; \boldsymbol{x} \in [0,1) \tag{2}$$

*Mirroring.* Smoothness usually is used to refer to bandlimited signals around the zero-frequency which can be represented by equation 1. However, we use the smooth finite-length signal to refer to a real-valued finite-length signal that has intuitively smooth behavior in its non-zero domain. figure 2 shows an example. The trigonometric functions in equation 2 can not approximate such signals well even if we shift and scale the domain to [0,1] because the original signal is not periodic. One possible solution to make the trigonometric functions still a good representative for the finite-length signal would be mirroring. figure 2 shows the shifted, scaled, and mirrored signals in 1D space.

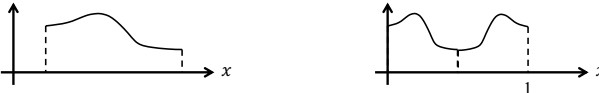

Figure 2: Mirroring, shifting and scaling.

*Extension to multi-dimensional real-valued mirrored signals.* equation 2 will be simplified for a real-valued $s$ just to include cosine terms. One can obtain the extension of equation 2 for real-valued mirrored signals as:

$$h(\boldsymbol{x}) = h(x_1, x_2, \ldots, x_d) = \sum_{m_1=0}^{M} \ldots \sum_{m_d=0}^{M} A_{m_1, m_2, \ldots, m_d} \cos\left(\pi(m_1 x_1 + \ldots + m_d x_d)\right), \tag{3}$$

where $\boldsymbol{x} \in [0,1]^d$ and $\mathbf{A}$ is a $d$-dimensional real tensor. To simplify the notation, we use $\boldsymbol{m}^T \boldsymbol{x}$ to refer $m_1 x_1 + \ldots + m_d x_d$ and $\boldsymbol{a}$ to refer vectorized $\mathbf{A}$. Starting from $m_1, m_2, \ldots, m_d$ all to be 0, one can put all the possible values of $\boldsymbol{m}$ as the columns of matrix $\boldsymbol{C} = [c]_{(M+1)^d \times d}$ with the same order as they appear in vectorizing of $\mathbf{A}$. Now we can rewrite equation 3 with matrix operations:

$$h(\boldsymbol{x}) = \sum_{k=1}^{(M+1)^d} a_k \cos\left(\pi \boldsymbol{C}_{k,:} \boldsymbol{x}\right) \tag{4}$$

*Vandermonde matrix.* Given $r$ non-uniform samples in $[0,1]^d$, the Fourier coefficients, $\boldsymbol{a}$, are the solution to the linear system of equations $h(x_i) = s_i, i = 1, 2, \ldots, r$. The Vandermonde matrix for this system is defined as:

$$\boldsymbol{V} = \cos\left(\boldsymbol{C}[x_1, x_2, \ldots, x_r]\right), \tag{5}$$

where the $\cos(.)$ is the element-wise cosine, and $[\ldots]$ shows the stacking operator of column vectors. So, the linear system of equations can be shortened by: $\boldsymbol{V}^T \boldsymbol{a} = \boldsymbol{s}$. Here $\boldsymbol{s}$ is the column vector of $r$ observed $s_i$ put together. In contrast to the 1D case, there is no simple theorem on the conditions to estimate $\boldsymbol{a}$ correctly. Roughly speaking, this needs the rank of $\boldsymbol{V}$ to be larger than the number of unknowns, i.e., $(M+1)^d$ or in other words, the number of samples ($r$) should be enough larger than $(M+1)^d$.

*Reconstruction of the preference function from the observed ratings.* We can state the problem of rating prediction, as the reconstruction of the preference function of each user ($f_u$) given the observed ratings of that user ($\mathcal{I}_u$). If we assign a d-dimensional characterization vector ($\boldsymbol{x}_i$) to each item $i$ that is assumed to lie in $\mathcal{X} = [0, 1]^d$, we can estimate the user $u$ Fourier coefficients as $\boldsymbol{a}_u = (\boldsymbol{V}_u^T)^\dagger \boldsymbol{s}_u$. At the starting point we do not know how items are distributed in $\mathcal{X}$ which means $\boldsymbol{V}_u$ will be inaccurate. So, optimizing the reconstruction loss gives fair characterstics for the items in $\mathcal{X}$:

$$\min_{\boldsymbol{x}_i, i \in \mathcal{I}} \sum_{u \in \mathcal{U}} \|\boldsymbol{V}_u^T (\boldsymbol{V}_u^T)^\dagger \boldsymbol{s}_u - \boldsymbol{s}_u\|^2. \tag{6}$$

# 3 LEARNING REPRESENTATIONS BY MINIMIZING RECONSTRUCTION LOSS

Minimizing equation 6, aside from the non-convexity of the cost function, implicitly involves solving $\boldsymbol{V}_u^T \boldsymbol{a}_u = \boldsymbol{s}_u$, which can in general be an ill-condition system of linear equations, specially when the user $u$ has few recorder ratings. To reliably estimate the Fourier coefficients $\boldsymbol{a}_u$ (user representations), group similar users into a number of clusters and use a single representative for each cluster (virtually increasing the number of available ratings). In addition, we further consider a Tikhonov ($L_2$) regularizer to improve the condition number. With this approach, we need to solve

$$\min_{\{\boldsymbol{x}_i, i \in \mathcal{I}\}, c} \mathcal{L} = \min \sum_{u \in \mathcal{U}} \|\boldsymbol{V}_{c(u)}^T \boldsymbol{a}_{c(u)} - \boldsymbol{s}_u\|^2 + \lambda \sum_{k \in \mathcal{C}} \|\boldsymbol{a}_k\|^2,$$
$$\text{s.t. } 0 \le \boldsymbol{x}_i < 1, \tag{7}$$

where $c : \mathcal{U} \to \mathcal{C}$ is the mapping of the users into clusters, $\mathcal{C}$ is the set of clusters and $\boldsymbol{V}_k$ is the Vandermonde matrix associated with the cluster $k$ (considering all the users in a cluster as a super-user). Hence, $\boldsymbol{V}_k$ is a function of $\{\boldsymbol{x}_i, i \in \cup_{\{u : c(u)=k\}} \mathcal{I}_u\}$. The penalty parameter $\lambda$ shall be tuned via cross validation. Moreover, the Fourier coefficients $\boldsymbol{a}_k$ for the cluster $k$ are obtained by:

$$\boldsymbol{a}_k = (\boldsymbol{V}_k \boldsymbol{V}_k^T + \lambda \boldsymbol{I})^{-1} \boldsymbol{V}_k \boldsymbol{s}_k, \tag{8}$$

where $\boldsymbol{s}_k$ is the vector of all observed ratings in the cluster $k$. In the sequel, we propose two approaches for minimizing equation 7. In the first approach (Section 3.1), we alternatively find $\min_{\{\boldsymbol{x}_i, i \in \mathcal{I}\}} \mathcal{L}$ and $\min_c \mathcal{L}$; as $\min_c \mathcal{L}$ requires a combinatorial search, we introduce an approximate algorithm, named *k-representation* (Section 3.1.1) inspired by the $k$-means technique. Each iteration of $k$-representation consists of assigning each user the cluster with the lowest reconstruction loss, and updating the cluster representatives. In the second approach (Section 3.2), we train a neural network to jointly characterize the items and cluster the users. For this, the loss function of equation 7 is modified to accommodate for soft clustering.

## 3.1 ALTERNATING OPTIMIZATION

### 3.1.1 $k$-REPRESENTATION FOR CLUSTERING THE USERS

The total loss in equation 7 can be divided into partial losses of the form

$$\mathcal{L}_k = \|\boldsymbol{V}_k^T \boldsymbol{a}_k - \boldsymbol{s}_k\|^2 \tag{9}$$

for each cluster $k$. We propose the *k-representation* (Algorithm 1) to minimize the overall cost iteratively. We first randomly set $\boldsymbol{a}_k$s, $k \in \mathcal{C}$; then, each user is assigned to a cluster $k$ for which its reconstruction loss (equation 7) is minimized. After dividing the users into clusters, the representative of each cluster is updated via equation 8, and we return again to the clustering task. Similar to the $k$-means, there is no theoretical guarantee that the method converges to the global optimizer; nevertheless, the overall loss is decreased in each iteration. We shall evaluate the performance of the method in the Section **??** on both synthetic and real data.

*Boosted k-representation.* By introducing both the clustering and $L_2$ regularization in equation 8, we have improved the robustness of the inverse problem. However, increasing the number of clusters is still a potential issue in estimating the cluster representatives. Here, we propose to learn an ensemble of weak binary clusterings instead of learning all clusters together (Algorithm 2). The idea is to find the residuals of predicted ratings for each user and fit a new clustering to the residuals. Due to the linearity of the prediction, the final representation is the sum of weak representations for each user.

---

**Algorithm 1** *k-representation*

---

**Input:**
    item characteristics $\{\boldsymbol{x}_i, i \in \mathcal{I}\}$
    available ratings by each user $\{\mathcal{I}_u, \boldsymbol{s}_u, u \in \mathcal{U}\}$
    number of clusters $|\mathcal{C}|$
    initialization variance of cluster representations $\sigma^2$
    $L_2$ penalty parameter $\lambda$.
**Output:**
    user clustering $c : \mathcal{U} \to \mathcal{C}$
    cluster representatives $\{\boldsymbol{a}_k, k \in \mathcal{C}\}$
**procedure** $k$-REPRESENTATION
    init. $\boldsymbol{a}_k$ from $\mathcal{N}(0, \sigma^2)$
    **for all** $k \in \mathcal{C}$ **do** Calculate $\boldsymbol{V}_k$ from $\boldsymbol{x}_i s$
    **repeat**
        **for all** $u \in \mathcal{U}$ **do** $c(u) \leftarrow \operatorname{argmin}_k \|\boldsymbol{V}_u^T \boldsymbol{a}_k - \boldsymbol{s}_u\|^2$
        **for all** $k \in \mathcal{C}$ **do** Update $\boldsymbol{a}_k$ via equation 8
    **until** convergence
    **return** $\{\boldsymbol{a}_k\}$ and $c$
**end procedure**

---

**Algorithm 2** *boosted k-representation*

---

**Input:**
    same as Algorithm 1
**Output:**
    user representatives $\{\boldsymbol{a}_u, u \in \mathcal{U}\}$
**procedure** BOOSTED $k$-REPRESENTATION
    **for all** $u \in \mathcal{U}$ **do** $\boldsymbol{a}_u \leftarrow 0$, , $\boldsymbol{s}_u^{(0)} \leftarrow \boldsymbol{s}_u$
    **for** $l = 1 : \lceil \log_2 |\mathcal{C}| \rceil$ **do**
        $\{\boldsymbol{a}_k^{(l)}\}, c^{(l)} \leftarrow k\text{-representation}(\{\boldsymbol{s}_u^{(l-1)}\})$
        **for all** $u \in \mathcal{U}$ **do** $\boldsymbol{a}_u^{(l)} \leftarrow \boldsymbol{a}_{c^{(l)}(u)}^{(l)}$,   $\boldsymbol{s}_u^{(l)} \leftarrow \boldsymbol{s}_u^{(l-1)} - \boldsymbol{V}_u^T \boldsymbol{a}_u^{(l)}$,   $\boldsymbol{a}_u \leftarrow \boldsymbol{a}_u + \boldsymbol{a}_u^{(l)}$
    **end for**
**end procedure**

---

### 3.1.2 OPTIMIZING ITEM CHARACTERISTICS

The second part of the alternating optimization is to minimize equation 7 w.r.t. item characteristics; i.e., $\{\boldsymbol{x}_i, i \in \mathcal{I}\}$, subject to $0 \leq \boldsymbol{x}_i < 1$. To simplify the equations, we rewrite the total loss in equation 7 with small modification as:

$$\mathcal{L} = \sum_{(u,i) \in \mathcal{O}^+} \rho(l_{u,i}) \tag{10}$$

where $l_{u,i}$ is $|S_{u,i} - \boldsymbol{v}_i^T \boldsymbol{a}_u|^2$ and $\mathcal{O}^+$ is the set of observed ratings. Here, $\rho$ is a saturating function that reduces the effect of outliers. In equation 7, it is simply the identity function; however, we chose $\rho(y) = 2(\sqrt{(1+y)} - 1)$ to better bound the unpredictable errors. We use the *Trust Region Reflective* (TRF) algorithm of the `scipy` python package as the optimization algorithm, which selects the updating steps adaptively and without supervision. To facilitate the optimization, we need to calculate the gradient of $l_{u,i}$ w.r.t. $\{\boldsymbol{x}_i\}$. It is obvious that $\nabla_{\boldsymbol{x}_j} l_{u,i}$ is zero for all $j \neq i$. For $j = i$ and $q = 1, ..., d$ (dimension of $\mathcal{X}$) we have:

$$\frac{\partial}{\partial x_{i,q}} l_{u,i} = 2\left(S_{u,i} - \boldsymbol{v}_i^T \boldsymbol{a}_u\right) \sum_{n=1}^{(M+1)^d} \pi a_{u,n} \sin(\pi \boldsymbol{C}_{n,:} \boldsymbol{x}_i) C_{n,q}. \tag{11}$$

*Pre-search*. Before using the TRF to optimize the loss, we make use of the current function evaluations to update item characteristics. Consider we are in the $t^{th}$ iteration and the values

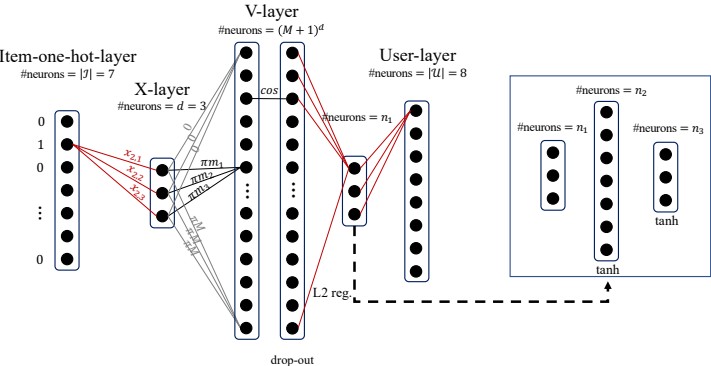

Figure 3: Neural network inspired from equation 13.

$\left\{ \boldsymbol{x}_i^{(t-1)}, \boldsymbol{V}_i^{(t-1)}, \boldsymbol{a}_i^{(t-1)} \right\}$ from the previous iteration are available. We define

$$\boldsymbol{x}_i^{(t-\frac{1}{2})} = \boldsymbol{x}_{\arg\min_{j \in \mathcal{I}} \| \boldsymbol{s}_i - \boldsymbol{V}_j^{(t-1)} \boldsymbol{a}_i^{(t-1)} \|^2}^{(t-1)}. \tag{12}$$

Then, we use $\left\{ \boldsymbol{x}_i^{(t-\frac{1}{2})} \right\}$ as the input to TRM and run a few iterations to get $\left\{ \boldsymbol{x}_i^{(t)} \right\}$. This pre-search makes the optimization process less prone to stagnate in local minima.

## 3.2 RECONST-NET: A FEED-FORWARD SHALLOW NETWORK FOR PREFERENCE RECONSTRUCTION

Recent advances in deep learning have made the neural networks great tools even for traditional well-known algebraic calculations. Specifically, neural networks can be optimized effectively with various methods and efficient implementations that boost training. Further, some useful techniques like batch normalization, drop-out, etc., are available, which helps to avoid overfitting. In this section, we will reformulate equation 7 for soft clusters and design an architecture that learns items' characteristics and users' representations concurrently. First we modify equation 7 for soft clustering. Consider $\boldsymbol{c} : \mathcal{U} \to \mathcal{C}^{|\mathcal{C}|}$ is the assignment function that determines how much each user is belonged to each cluster. We intentionally do not constraint norm of the $\boldsymbol{c}$ and let it to be scaled appropriately for different user. The total reconstruction loss is:

$$\mathcal{L} = \sum_{u \in \mathcal{U}} \| (\sum_{k \in \mathcal{C}} c_k(u) \boldsymbol{V}_k^T \boldsymbol{a}_k) - \boldsymbol{s}_u \|^2 + \lambda \sum_{k \in \mathcal{C}} \| \boldsymbol{a}_k \|^2 \tag{13}$$

figure 3 shows an inspired architecture from 13. It consists of six layers (four hiddens) which three of them are trainable. The observed ratings will be supplied per item. Each neuron corresponds to an item at the input layer, and the input data should be one-hot vectors. The next layer is *X-layer*, which is a dense layer with $d$ units. We interpret the weights from unit $i$ in the input layer to the X-layer as $\boldsymbol{x}_i$. At the *V-layer*, the item's representation ($\boldsymbol{x}_i$) is multiplied by $\boldsymbol{C}$ (EQ) and forms $\boldsymbol{v}_i$. V-layer does not have any trainable parameters. Next, there is the soft-clustering layer, a dense layer with $n_1$ neurons. We interpret weights going from the V-layer to unit $k$ of the soft-clustering layer as $\boldsymbol{a}_k$, i.e., the representation of the $k^{th}$ soft cluster. Finally, the output layer or equivalently the *User-layer* is a dense layer with $|\mathcal{U}|$ units. Weights going from the soft-clustering layer to unit $u$ determine $\boldsymbol{c}(u)$.

The drop-out layer (not depicted) after V-layer has an important role in preventing the network from overfitting. Dropping-out makes sense because the observed ratings usually come with a lot of uncertainty in a real application, i.e., the same user might rate the same item differently when asked for re-rating, and this is the nature of real data. The drop-out layer prevents overfitting by stopping the network from relying on a specific part of the items' characteristics.

One way to increase the capacity of the method and capture non-linear interactions is to let the user's representation be a nonlinear function of clusters' reconstructed ratings. i.e., to change equation 13

Table 1: Datasets summary

| DATASET | #USER | #ITEM | #RATINGS | #DENSITY | RATING RANGE |
|---|---|---|---|---|---|
| Synthetic | 50 | 200 | variable | variable | 1, 2, ..., 5 |
| ML-100k | 943 | 1,682 | 100,000 | 0.063 | 1, 2, ..., 5 |
| ML-1M | 6,040 | 3,706 | 1,000,209 | 0.045 | 1, 2, ..., 5 |

Table 2: Training settings

| METHOD | DATASET | $d$ | $M$ | $\lambda$ | OTHERS |
|---|---|---|---|---|---|
| Alternating optimization 3.1 | ML-100k | 3 | 4 | 0.1 | boosted k-rep. with 4 learners |
| | ML-1M | 3 | 4 | 0 | boosted k-rep. with 5 learners |
| RECONST-NET 3.2 | ML-100k | 3 | 10 | 10 | $n_1 = 10, n_2 = 100, n_3 = 10$, drop-out= 0.1 |
| | ML-1M | 4 | 10 | 10 | $n_1 = 15, n_2 = 100, n_3 = 15$, drop-out= 0.1 |

as:

$$\mathcal{L} = \sum_{u \in \mathcal{U}} \| \sum_{k \in \mathcal{C}} c_k(u) g(\{\boldsymbol{V}_q^T \boldsymbol{a}_q\}) - \boldsymbol{s}_u \|^2 + \lambda \sum_{k \in \mathcal{C}} \|\boldsymbol{a}_k\|^2 \tag{14}$$

here $g$ can be an arbitrary non-linear function. In figure 3 we have proposed $g$ as two additional hidden layers in right panel with $tanh$ activation. Here, we usually choose $n_1 = n_3$ and equals our expectation from the number of soft clusters in the data. Still one can interpret the last hidden layer as the soft clustering layer.

### 3.3 COMBINING MULTI PREDICTORS

*Combining user-based and item-based methods.* Till now, we have assumed that items are mapped to a vector space, and users have preference functions (user-based method), but there is no reason not to consider it reversely (item-based method). A simple way of combining user-based and item-based methods is to do a linear regression from each method's output to the observed ratings. The validation part of the data will be used for estimating the coefficients of regression. We will see that combining user-based and item-based methods significantly improve the prediction of test data.
*Leveraging ensemble of predictors.* A more complicated but effective way of combining is to leverage an ensemble of combined user-based and item-based methods. Consider we have a predictor $f^{(t)}(\boldsymbol{S})$ at iteration $t$ that predicts observed ratings $\boldsymbol{S}$ for training the model. At each iteration, we calculate the residuals of the predicted ratings and pass it to the next predictor: $\boldsymbol{S}^{(t+1)} = \boldsymbol{S}^{(t)} - f^{(t)}(\boldsymbol{S}^{(t)})$. The next predictor, $f^{(t+1)}$, uses $\boldsymbol{S}^{(t+1)}$ for training its model. The final predictor leveraged from $f^{(t)}, t = 1, 2, ..., T$, uses the sum of all predictions: $f(\boldsymbol{S}) = \sum_{t=1}^{T} f^{(t)}(\boldsymbol{S})$.

## 4 EXPERIMENTS

To evaluate the proposed methods, we use three datasets: a synthetic dataset besides the well known ML-100k and ML-1M datasets (Harper & Konstan, 2015). The details of each dataset can be found in Table 1. The synthetic data is created to assess our clustering methods; for this purpose, a number of cluster representatives are randomly chosen in the low-frequncy domain (Guaranteed to be smooth) and then, each user is placed randomly around one of the representatives. The distance of the users to the associated representative (within-cluster variance) is varied in different tests. Look at the Appendix for detailed discussion.

The MovieLens datasets[1] ML-100k and ML-1M are two of the most common benchmarks for collaborative filtering. For the ML-100k, we follow the pre-specified train-test split; i.e., 75%, 5%, and 20% of the total available data is used for training, validation, and test, respectively. For ML-1M, these numbers are 90%, 5%, and 5%, respectively.

---

[1] https://grouplens.org/datasets/movielens/

Table 3: RMSE comparison

| METHOD | ML-100k | ML-1M |
|---|---|---|
| PMF | 0.952 | 0.883 |
| GMC (Kalofolias et al., 2014) | 0.996 | - |
| Factorized EAE (Hartford et al., 2018) | 0.920 | 0.860 |
| IGMC (Zhang & Chen, 2019) | **0.905** | 0.857 |
| GC-MC (Berg et al., 2017) | 0.910 | 0.832 |
| Bayesian timeSVD++ (Rendle et al., 2019) | 0.886 | **0.816** |
| sRGCNN (Monti et al., 2017) | 0.929 | - |
| GRALS (Rao et al., 2015) | 0.945 | - |
| Alternating optim. (ours) | 0.920 | 0.860 |
| RECONST-NET (ours) | 0.909 | 0.862 |

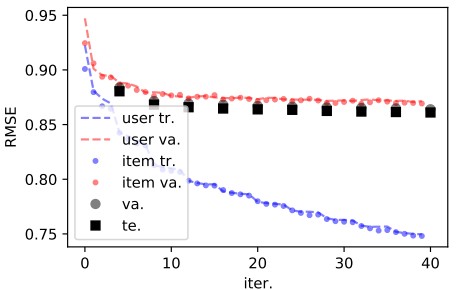

(a) Alternating optimization in ML-1M

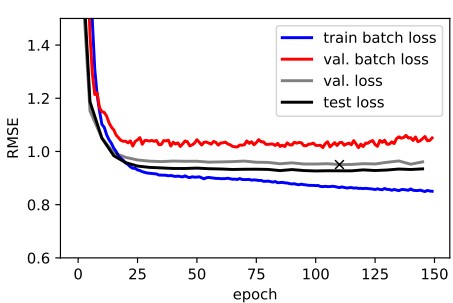

(b) RECONST-NET training with ML-100k

Figure 4: Training process for different methods

### 4.1 PREDICTION EVALUATION

*Training process.* The training settings of both proposed methods, alternating optimization, and RECONST-NET for each dataset are provided in Table 2. Figure 4 shows the RMSE on two different datasets during the training stage. The validation is reserved for parameter tuning, and performance on test data is reported for both methods. Figure 4a clearly reveals the performance gain achieved by combining the user- and item-based techniques. Further, the stair shape decrease of the training loss (and validation loss) confirms the suitability of leveraging the ensemble of predictors.

*Performance comparison.* We further conduct experiments on ML-100k and ML-1M datasets. We have ignored the available side information in both datasets. Therefore, for the performance comparison in Table 3, we have included only methods that do not take into account these side information. Although neither of the alternating optimization and RECONST-NET record the best RMSE, they yield very good results despite their simplicity and interpretability.

## 5 CONCLUSION

In this article, we formulated the rating prediction problem as a reconstruction problem with a smoothness assumption. The proposed methods are all simple and interpretable but show significant performance comparing to state-of-the-art methods. Specifically, we interpreted different layers of the designed network and evaluated our interpretation in synthetic design. The proposed architecture and rich frequency-domain feature can be a basis for future research on interpretable recommender systems.

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

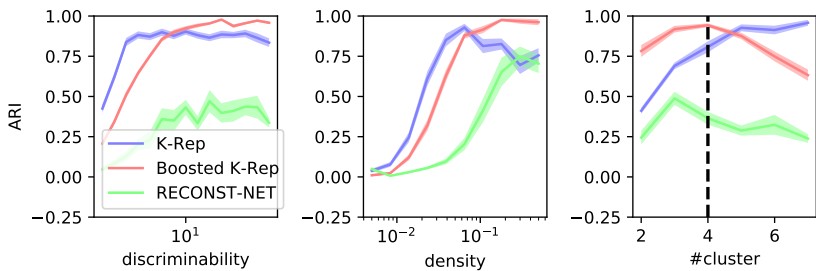

Figure 5: The performance of the clustering techniques on synthetic data

# A APPENDIX

## A.1 CLUSTERING EVALUATION

In Section 3.1.1, we proposed the $k$-representation clustering and its boosted version. Here, we study their performances via experiments on synthetic data. We recall that the mentioned methods do not explicitly penalize miss-clustering; instead, they minimize the within-cluster reconstruction loss. As a result, we expect these methods to perform fairly well when the clusters are distinguishable. To measure the matching between the identified clusters and the original ones, we employ the *Adjusted Rank Index* (ARI) (look at Vinh et al. (2010)). For two clusterings $c_1$ and $c_2$ with the same domain $\mathcal{U}$, if we form the contingency matrix $\boldsymbol{N}$ with the $(i,j)$ element as $|\{u, c_1(u) = i, c_2(u) = j\}|$, then, ARI defined based on the elements, column and row sums of $\boldsymbol{N}$, intuitively shows the rate of agreement between the two clusterings if $u$ is randomly chosen from $\mathcal{U}$. It takes the maximum value $1$ for identical clusterings and the minimum value $0$ when the clusterings are perceived as fully random with respect to each other. As explained, ARI has the advantage of comparing two clusterings even with different number of clusters.

To use the ARI metric, we need a hard clustering of the users. For this purpose, we associate each user in our User-layer (soft clustering layer) in Figure 3 to the neuron (cluster) in the last hidden unit with the largest absolute weight. Although this technique violates the main goal in soft clustering, it provides us with a measure of clustering accuracy.
In Figure 5, the performance of $k$-representation (1), boosted $k$-representation (2) and modified results from the soft clustering layer (Figure 3) are depicted for three scenarios. As expected, we obtain inferior result from the soft-clustering layer; however, as ARI is above zero, the clustering of this layer is not irrelevant. In the left plot in Figure 5, the ARI curves in terms of the discrimination index of the clusters (the ratio of between-cluster to within-cluster variances) are presented. We observe that the boosted $k$-representation works better in cases with higher discrimination index, but loses its performance in case of cluttered clusters.
The center plot in Figure 5, shows ARI changes by varying the density in the rating matrix (ratio of the number of observed to non-observed entries). While the $k$-representation has the best performance at low densities, its performance drops when the density exceeds a threshold; this might be due to the involved regularization term. Finally, in the right plot of Figure 5, we have changed the number of clusters in all methods; the correct number of clusters is kept fixed at $4$. As we see in these plots, none of the $k$-representation and its boosted version dominate the other one in all regimes.

