# OpenReview forum: "Collaborative Filtering with Smooth Reconstruction of the Preference Function"
_ICLR.cc/2021/Conference — Reject_

### Official Review · AnonReviewer3 · 2020-10-21
**This paper seems to be a preliminary work and should not be considered for 2nd round review.**

**Rating:** 3
**Confidence:** 5

**Review:**

The paper is not well-written, so it is hard for me to fully understand the whole paper. It seems to me that the authors tried to model user preferences with smooth functions for collaborative filtering and results on two Movielens datasets showed that the proposed method is comparable to several baseline methods.

Pros:
1.	The idea of capturing frequency domain information for collaborative filtering seems interesting to me, although I did not quite understand how this information is obtained in this work.
2.	The authors mentioned that their method can be interpretable, which may be more interesting if they can provide some case studies for explaining recommendations using the proposed method.

Cons:
1.	The motivation is not clear to me, i.e., why smooth functions are important in modelling user preferences in collaborative filtering is not well explained.
2.	The authors seem to have a narrow understanding of collaborative filtering. For instance, the authors mentioned that “the utilization of deep architectures is limited to shallow networks” in exiting CF literature. However, there are a lot of recent works building upon state-of-the-art deep learning techniques as far as I know.
3.	The idea of this paper is very similar to a recent work (Harald Steck, Embarrassingly Shallow Autoencoders for Sparse Data, WWW ’19). The main difference may be that this work adopted clustering of users instead of individual user when modelling user interests.
4.	The experimental results are not encouraging. The proposed method is actually much worse than Bayesian TimeSVD++ (Rendle et al., 2019) and IGMC (Zhang & Chen, 2019). So, it is hard to understand why this alternative method is promising in the area.
5.	The presentation could be improved. The authors mentioned that their code has been published but no link was provided in the paper.

Overall, I think this paper is a preliminary work and is not ready for publication. I think the authors should revise the paper and submit the revised version to another conference.

---

### Official Review · AnonReviewer2 · 2020-10-22
**artful idea, but not very clear**

**Rating:** 4
**Confidence:** 3

**Review:**

The paper has proposed a smooth reconstruction of the preference function in collaborative filtering for recommender system. The evaluations have generally demonstrate the proposed method. However, there are several concerns left.

1, The paper has proposed utilizing interpretable layers to implicitly cluster the users. However, it is not clear how the interpretable layer measure the interpretability using qualitative metric. I don't believe the the weights appeared in the X layers gains the interpretability to tell the model how the users may be clustered. In terms of the contribution of the interpretable layer, there should be ablation study to demonstrate the effectiveness of the interpretable layers(without x-layer).

2, The experiments mostly look good. However, they are not superior compared to the baselines. It is not required to outperform the baseline results that listed in Table 3, but necessary analysis should be conduct why the gap exists. If it is claimed that some baselines use side information, it is vital to eliminate the side informations so that to make a fair experimental comparisons.

3, The experiments could be explored to other datasets to show the generalization capability of the proposed method. Even in Movie-lens dataset, if would be easier to extend to ranking based evaluation(using NDCG, F1 etc), rather than purely rate-based evaluations since both of them are practical metric in recommender system.

4, Some procedure of the proposed methods are not intuitive, e.g.In Alg 2, why take log2|C| as the total size of k-representations?

5. some typos and gammer errors. e.g.  page 4, 'in the section ??'   page 5,  'which selects the updating steps adaptively and without supervision.

---

### Official Review · AnonReviewer4 · 2020-10-23
**Ignores key problem that ratings are missing not at random**

**Rating:** 3
**Confidence:** 4

**Review:**

This paper proposes an approach based on Fourier transforms to predict ratings in collaborative filtering problems. The paper’s scope (“smooth reconstruction functions”) gets immediately narrowed down to Fourier transforms--it would be nice to provide some motivation for this choice over alternative smooth functions. The paper then clusters the users as a way to reduce the number of parameters in the model, given that the Fourier transform itself does not reduce it. As a further step, the clustering is replaced by a soft-clustering learned by a neural network. In the experiments, the RMSE of the rating prediction problem is worse than some baselines and better than others.

Besides these technical steps, from a more big-picture perspective, I am not sure if the problem of rating prediction as cast in this paper, misses a key point. The key point I am concerned about is that the observed ratings are missing not at random [a]. For this reason, the collaborative-filtering literature abandoned the minimization of RMSE on the OBSERVED ratings ten years ago. Two different avenues have been pursued since then: most of the papers switched to ranking the entire catalog of items, e.g, see [b] to get started. A few papers continued with rating prediction, but stated the problem correctly by taking into account the fact that the ratings are missing not at random, eg., [c,d].

In this submission, the problem statement at the top of page 4, and Eq. 6, was not clear to me: while s was defined clearly in the Fourier transform earlier in the paper,  I did not find a definition of s_u in Eq 6 in the context of rating prediction, i.e., is this the vector of ratings of user u? Only the observed ratings? How are the unobserved/missing ratings of user u treated in the proposed approach?

Given the RMSE-values in the experiments, my best guess is that the model was trained on the observed ratings only, ignoring the key problem that the ratings are missing not at random.

I feel like a rating-prediction paper that ignores the key problem of collaborative filtering, i.e., the fact that ratings are missing not at random cannot be accepted (and should actually be desk-rejected).

I encourage the authors to modify the approach to account for this key problem of collaborative filtering. Alternatively, this approach may be useful for different applications, like compressive sensing problems where the observations are truly random.

[a] Collaborative Prediction and Ranking with Non-Random Missing Data
by B. Marlin and R. Zemel (RecSys 2009 Best Paper)

[b] Training and testing of recommender systems on data missing not at random
by H. Steck (KDD 2010)

[c] Probabilistic Matrix Factorization with Non-random Missing Data
by J.M. Hernández-Lobato, N. Houlsby, and Z. Ghahramani (ICML 2014)

[d] Modeling User Exposure in Recommendation
by D. Liang et al. (WebConf 2016)

---

### Official Review · AnonReviewer1 · 2020-10-26
**Collaborative Filtering with Smooth Reconstruction of the Preference Function**

**Rating:** 4
**Confidence:** 4

**Review:**

Summary：

In this paper, the authors regarded the rating prediction as a reconstruction problem with a smoothness assumption. They proposed two approaches for smoothly reconstructing the preference function of users and conducted experiments on both a synthetic dataset and two benchmark datasets (ml-100k and ml-1m). However, there are still some technical issues in this article, such as unconvincing experiments and confusing descriptions. As a result, I suggest the paper should be rejected.

Detailed Comments：

With respect to the problem of rating predictions in collaborative filtering, this work proposed to estimate the predictions by assuming an underlying smooth preference function for each user, the quantization at each given vector yields the associated rating. Then, they developed two methods including k-representation motivated by k-means and reconst-net, a feed-forward neural network, to do this. The authors also conducted experiments on two benchmark datasets (ml-100k and ml-1m) to test the performance of proposed algorithms.

The key strengths:
- The perspective of reconstructing the user preference functions is novel.
- They proposed two effective methods to estimate the preference functions.
- An alternating optimization method is proposed that effectively optimizes a non-convex loss function and extracts user and item representations.

Reason to Reject:

- The contribution of “the reconstruction of user preference functions” is overclaimed. Much literature in the area of CF regards the rating prediction problem as a reconstruction problem, such as Matrix Factorization (MF) based algorithms [1,2]. The authors should include these methods and elaborate on the difference between them.
- In the introduction, the authors pointed out that the contemporary deep network in RS is merely limited to shallow networks and they addressed this problem in this work. However, the reconst-net only involves three trainable layers. It is not clear how do you solve the “shallow” problem?
- The experiments are not convincing. The authors claimed that “their performance on benchmark datasets is remarkable”, but in experiments, we see that the performance of the proposed model is not competitive, e.g., the performance gap on ml-1m is about 5%. In addition, the authors did not dive into these results and analyze them. They should also include some MF-based methods for rating prediction problem, such as [1,2]
- Many details and explanations are missing and confusing. For example, the intuitions of figure 1 and figure 2 are not clear, and the authors should give more explanations with respect to these figures. In figure 4a, what does the legend represent, such as “user tr.”, “va.” and “item tr.”?
- some typos:
  - “in the Section ?? on both synthetic and real data”
  - “figure 2 shows an example”，” figure” should be “Figure”

[1] FISM: factored item similarity models for top-N recommender systems. KDD’13
[2] Factorization meets the neighborhood: a multifaceted collaborative filtering model. KDD’08

---

### Decision · Program_Chairs · 2021-01-07
**Final Decision**

**Decision:**

Reject

**Comment:**

This paper mostly received negative scores. A few reviewers pointed out that the idea of modeling user preference in the frequency domain seems novel and interesting. However, there are a few concerns around the clarity of the paper, the motivation of the proposed approach, as well as the experimental results being unconvincing (both in terms of execution as well as exploration of the results). The authors did not provide a response. Therefore, I recommend reject.